# Modeling the Strain-Range Dependent Cyclic Hardening of SS304 and 08Ch18N10T Stainless Steel with a Memory Surface

**Radim Halama** [1],\*, **Jaromír Fumfera** [2], **Petr Gál** [1,3], **Tadbhagya Kumar** [4] and **Alexandros Markopoulos** [1]

[1]  Department of Applied Mechanics, Faculty of Mechanical Engineering, VSB-Technical University of Ostrava, 17.listopadu 2172/15, 70800 Ostrava, Czech Republic
[2]  Department of Mechanics, Biomechanics and Mechatronics, Faculty of Mechanical Engineering, Center of Advanced Aerospace Technology, Czech Technical University in Prague, Technicka Street 4, 16607 Prague 6, Czech Republic
[3]  ÚJV Řež, a. s., Hlavní 130, Řež, 250 68 Husinec, Czech Republic
[4]  Mechanical and Aerospace Engineering Department, University of Florida, Gainesville, FL 32611, USA
\*  Correspondence: radim.halama@vsb.cz; Tel.: +420-597-321-288; Fax: +420-596-916-490

**Abstract:** This paper deals with the development of a cyclic plasticity model suitable for predicting the strain range-dependent behavior of austenitic steels. The proposed cyclic plasticity model uses the virtual back-stress variable corresponding to a cyclically-stable material under strain control. This new internal variable is defined by means of a memory surface introduced in the stress space. The linear isotropic hardening rule is also superposed. First, the proposed model was validated on experimental data published for the SS304 material (Kang et al., Constitutive modeling of strain range dependent cyclic hardening. Int J Plast 19 (2003) 1801–1819). Subsequently, the proposed cyclic plasticity model was applied to our own experimental data from uniaxial tests realized on 08Ch18N10T at room temperature. The new cyclic plasticity model can be calibrated by the relatively simple fitting procedure that is described in the paper. A comparison between the results of a numerical simulation and the results of real experiments demonstrates the robustness of the proposed approach.

**Keywords:** cyclic plasticity; cyclic hardening; finite element method; austenitic steel 08Ch18N10T; stainless steel 304

## 1. Introduction

The SS304 material, which includes 18 percent chromium and eight percent nickel, is the most widely-used austenitic stainless steel. It has good drawability and welding properties together with strong corrosion resistance. Austenitic stainless steel 08Ch18N10T is a chrome-nickel steel that is stabilized by titanium. This steel is widely used in the nuclear industry for piping systems and reactor internals in the Russian-designed VVERwater-water power reactors for nuclear power plants (NPP). Reactor internals are the part of an NPP that provides support, guidance, and protection for the reactor core and for the control elements. The block of guided tubes, the core barrel, the core barrel bottom, and the core shroud are some of the internal components that are exposed to very harsh operating regimes. The operating regime, e.g., heating and shut-downs, has a significant influence on the service life of the components. The vibration and pressure pulsation of the water pumps also have to be taken into account. These regimes expose the reactor internals to cyclic loading.

In practice, cyclic loading of structural parts can lead to the formation and propagation of cracks through the process referred to as fatigue. In all areas of industry, the operational safety of machinery

depends on an appropriate design process, which includes an analysis of all possible critical states. In the low-cycle fatigue domain, seismic analysis and the simulation of operational tests of the piping systems of NPPs may be used as an example. In these cases, it is crucial to have an accurate description of the stress–strain behavior of the material that is being considered.

Phenomenological models [1] are the most widely-used models in practical applications. Their goal is to provide an as accurate as possible description of the stress–strain behavior of the material, which is found on the basis of experiments [2]. The stress–strain behavior of structural materials under cyclic loading is very diverse, and a case-by-case approach is required [3].

The most progressive group of cyclic plasticity models, which are commonly encountered in commercial finite element method programs, is the single yield surface models based on differential equations. Their development is closely linked to the creation of a nonlinear kinematic hardening rule with a memory term, introduced by Armstrong and Frederick in 1966 for the evolution of back-stress [4] and the discovery by Chaboche [5] of the vast possibilities offered by the superposition of several back-stress parts.

Developments in the field of non-linear kinematic hardening rules were mapped in detail in [1]. In the current paper, we will mention only the most important theories. In 1993, Ohno and Wang [6] proposed two nonlinear kinematic hardening rules. For both models, it was considered that each part of the back-stress had a certain critical state of dynamic recovery. Ohno–Wang Model I leads to plastic shakedown under uniaxial loading with a nonzero mean axial stress value (no ratcheting), and under multiaxial loading, it gives lower accumulated plastic deformation values than have been observed in experiments. The memory term of Ohno–Wang Model II [6] is partially active before reaching the critical state of dynamic recovery, which allows a good prediction of ratcheting under uniaxial loading and also under multiaxial loading. The Abdel-Karim–Ohno nonlinear kinematic hardening rule [7] was published in 2000. This rule is in fact a superposition of the Ohno–Wang I and Armstrong–Frederick rules. The proposed model was designed to predict the behavior of materials that exhibit a constant increment of plastic deformation during ratcheting. Other modifications to this kinematic hardening rule, leading to a better prediction of uniaxial ratcheting and also multiaxial ratcheting, were proposed by one of the authors of the paper [8]. In order to capture the additional effects of cyclic plasticity, the concept of kinematic and isotropic hardening has been further modified. Basically, the available theories can be divided into two approaches. The first approach is related to the actual distortion of the yield surface [9–11], while the second approach is related to the memory effect of the material [5,12]. The effect of cyclic hardening as a function of the size of the strain amplitude is usually assumed in the second approach.

The first comprehensive model of cyclic plasticity with a memory surface was proposed by Chaboche and co-authors in [5]. Chaboche's memory surface was established in the principal plastic strain space and captures the influence of plastic strain amplitude and also the mean value of the plastic strain. The memory surface is associated with a non-hardening strain region in a material point, as was explained by Ohno [12] for the general case of variable amplitude loading. Memory surfaces established in the stress space have also been developed. Their main advantage is that they enable more accurate ratcheting strain prediction to be achieved, as presented by Jiang and Sehitoglu in their robust cyclic plasticity model [13].

It should be mentioned that both of these memory surface concepts lead to an increase in the number of material parameters and in the number of evolution equations, which complicates their use in engineering practice.

The original application of the memory surface, introduced by Jiang and Sehitoglu, was extended by some authors of the present paper to capture the memory effect of ST52 material, in [14]. Uniaxial experiments indicated that, in the case of a cyclically-softening/hardening material, larger strain amplitudes cause a significant change in the shape of the hysteresis loops. Only a very small number of researchers in the field of cyclic plasticity have investigated the influence of strain amplitude

on the cyclic hardening effect. Good agreement with experiments has been achieved in the case of steel SS304 [15], but at the cost of defining more than 70 material parameters.

Some of the material models have been used to capture cyclic material behavior. To describe the cyclic behavior of SAE4150 martensitic steel [16], Schäfer et al. considered three kinematic hardening models, i.e., the Chaboche [5], Armstrong-Frederick [4] and Ohno–Wang [6] models. They used these kinematic approaches to simulate the micromechanical behavior of the selected material. Moeini et al. [17] used the Chaboche model [5] to predict the low cyclic behavior of dual-phase steel. The selected kinematic hardening model provided good agreement with experimental results. Msolli used the unified viscoplastic model [18] developed by Chaboche when modeling the elastoviscoplastic behavior of JLF-1 steel at higher temperatures (400 °C and 600 °C). In this study, the Chaboche model was slightly modified to capture cyclic hardening followed by cyclic softening. The material model also falls into the category of coupled damage models. The material model showed good agreement with the experimental results. The effect of torsional pre-strain on low cycle fatigue performance of SS304 was studied in [19]. Kang et al. [20] used the viscoplastic constitutive model with the extended Abdel-Karim–Ohno nonlinear kinematic hardening rule with some temperature-dependent terms. This constitutive model was verified on uniaxial and non-proportional multiaxial ratcheting experimental results at room temperature and at elevated temperatures. Another viscoplastic constitutive model was used by Kang, Gao, and Yang [21] in their study to simulate uniaxial and multiaxial ratcheting of cyclically-hardening materials. They used the Ohno–Wang kinematic hardening rule with the critical state of dynamic recovery. The effect of loading history was also considered by introducing a fading memorization function for the maximum plastic strain amplitude.

This paper shows the advantages of using the memory surface established by Jiang and Sehitoglu in 1996 [13] to treat the impact of the strain amplitude on the material stress response. The new theory is shown on the kinematic hardening rule based on Chaboche's model with three back-stress parts, but it can also easily be applied to the Abdel-Karim–Ohno model or its modified version with promised ratcheting prediction [8]. Recently, an approach was introduced that takes into account a new internal variable referred to as virtual back-stress, corresponding to a cyclically-stable material. This provides an easy way to identify the parameters and to use fewer material parameters than in earlier models, for example [21]. New experimental results from uniaxial fatigue tests realized on 08Ch18N10T at room temperature are presented and subsequently used for the validation of the new cyclic plasticity model.

## 2. New Constitutive Model

In this paper, isothermal conditions are considered, and the influence of the strain rate is neglected. However, the model can be extended by standard techniques for use in the area of viscoplasticity [7].

### 2.1. Yield Surface and Flow Rule

In this work, the concept of a single yield surface for metallic materials is used, based on the von Mises yield function, which can be expressed for the general mixed hardening model as:

$$f = \sqrt{\frac{2}{3}(\boldsymbol{s} - \boldsymbol{a}) : (\boldsymbol{s} - \boldsymbol{a})} - Y = 0, \tag{1}$$

where $\boldsymbol{s}$ is the deviatoric part of stress tensor $\boldsymbol{\sigma}$, $\boldsymbol{a}$ is the deviatoric part of back-stress $\boldsymbol{\alpha}$, and the current size of yield surface (or the actual yield stress) $Y$ is defined as the sum of the isotropic variable $R$ and the initial size of the yield surface $\sigma_y$ (the yield strength) by the equation:

$$Y = \sigma_y + R. \tag{2}$$

It should be mentioned that the colon between the second-order tensors in Equation (1) denotes their inner product $x : y = x_{ij}y_{ij}$ (considering the Einstein summation convention).

The associative plasticity is considered, so the normality flow rule is considered in the case of active loading:

$$d\epsilon^p = d\lambda \frac{\partial f}{\partial \sigma}. \tag{3}$$

This expresses mathematically that the plastic strain increment $d\epsilon^p$ is collinear with the exterior normal to the yield surface for the current stress state. In associative plasticity, the scalar multiplier $d\lambda$ is equal to the accumulated plastic strain increment $dp$, which is defined as:

$$dp = \sqrt{\frac{2}{3}d\epsilon^p : d\epsilon^p}. \tag{4}$$

### 2.2. Virtual Back-Stress

A new internal variable is established to provide an easy way to calibrate the model. The variable is the back-stress of a cyclically-stable material corresponding to the response of the material investigated under a large strain range. It will be referred to as the virtual back-stress. The Chaboche superposition of the back stress parts is used in the following form:

$$\alpha_{virt} = \sum_{i=1}^{M} \alpha_{virt}^i \tag{5}$$

taking into consideration the nonlinear kinematic hardening rule of Armstrong and Frederick [4] for each part:

$$d\alpha_{virt}^i = \frac{2}{3}C_i d\epsilon_p - \gamma_i \alpha_{virt}^i dp, \tag{6}$$

where $C_i$ and $\gamma_i$ are material parameters. For all calculations in this paper, the superposition of three kinematic hardening rules ($M = 3$) will be used.

It should be mentioned that the virtual back-stress is used only in the definition of the memory surface, which will be described in the next section. Zero components of the virtual back-stress are considered in the initial state. The increment of the virtual back-stress is calculated according to Equations (5) and (6) assuming the current increment of accumulated plastic strain $dp$ and the current increment of plastic strain tensor $d\epsilon_p$ in each iteration of the local problem. Further details of the implementation algorithm that is used can be found in [22], where a more complex model with the memory surface of Jiang and Sehitoglu [13] was considered.

### 2.3. Memory Surface

To provide a correct description of the cyclic hardening for various strain ranges, a memory surface in the stress space is established. The concept is analogous to the theory of Jiang and Sehitoglu [13]. A scalar function is introduced to represent the memory surface in the deviatoric stress space:

$$g = \|\alpha_{virt}\| - R_M \leq 0, \tag{7}$$

where $R_M$ is the size of the memory surface and $\|\alpha_{virt}\|$ is the magnitude of the total virtual back-stress, which is defined as $\|\alpha_{virt}\| = \sqrt{\alpha_{virt} : \alpha_{virt}}$. The evolution equation ensuring the possibility of memory surface expansion, Figure 1, is therefore:

$$dR_M = H(g) \langle L : d\alpha_{virt} \rangle, \tag{8}$$

where

$$L = \frac{\alpha_{virt}}{\|\alpha_{virt}\|}.$$

(9)

Contraction of the memory surface is not allowed in this paper. It can be implemented according to the stress space-based memory surface concept of Jiang and Sehitoglu [13].

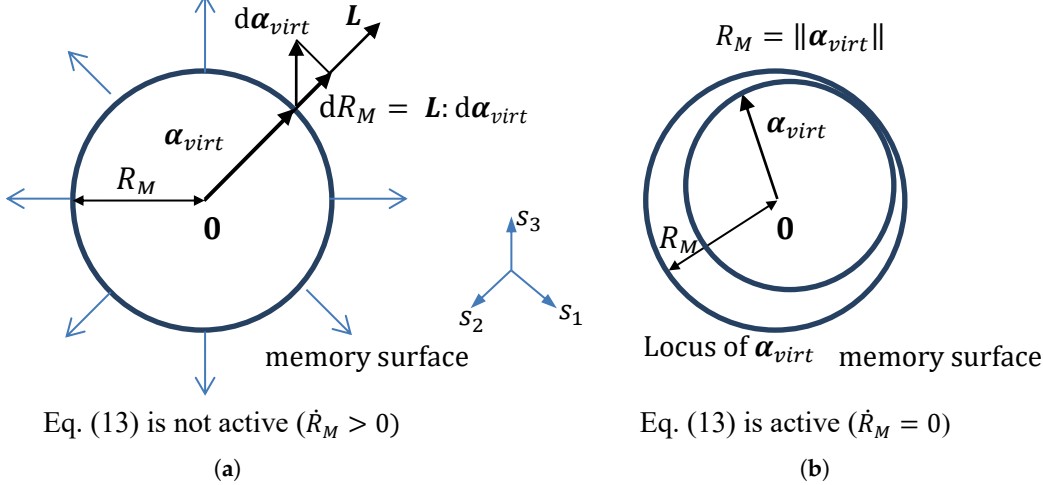

**Figure 1.** Expansion of the memory surface and the stabilized memory surface. (**a**) Equation (13) is not active; (**b**) Equation (13) is active.

### 2.4. Kinematic Hardening Rule

Consistent with the previous sections, the back-stress is composed of $M$ parts:

$$\alpha = \sum_{i=1}^{M} \alpha_i,$$

(10)

but the memory term is dependent on the size of memory surface $R_M$ and accumulated plastic strain $p$; thus:

$$d\alpha_i = \frac{2}{3} C_i d\epsilon_p - \gamma_i \phi(p, R_M) \alpha_i dp,$$

(11)

where $C_i$ and $\gamma_i$ are the same as in Equation (6). The multiplier $\phi$ of parameters $\gamma_i$ is composed of a static part and a cyclic part:

$$\phi(p, R_M) = \phi_0 + \phi_{cyc}(p, R_M),$$

(12)

where $\phi_0$ has the meaning of a material parameter, while the cyclic part is variable and can change only in the case of $\dot{R}_M = 0$. In this case, the evolution equation is defined in the following way:

$$d\phi_{cyc} = \omega(R_M) \cdot (\phi_\infty + \phi_{cyc}(p, R_M)) \, dp.$$

(13)

$$\phi_\infty(R_M) = A_\infty R_M^4 + B_\infty R_M^3 + C_\infty R_M^2 + D_\infty R_M + F_\infty,$$

(14)

$$\omega(R_M) = A_\omega + B_\omega R_M^{-C_\omega} \text{ for } R_M \geq R_{M\omega},$$

(15)

$$\omega(R_M) = A_\omega + B_\omega R_{M\omega}^{-C_\omega} \text{ otherwise,} \tag{16}$$

where $A_\infty$, $B_\infty$, $C_\infty$, $D_\infty$, $F_\infty$, $A_\omega$, $B_\omega$, $C_\omega$, $R_{M\omega}$, and $R_{M0}$ are additional parameters to Chaboche's material parameters $C_i$ and $\gamma_i$. The evolution parameter $\omega$ directs the rate of cyclic hardening behavior according to the current size of memory surface $R_M$.

*2.5. Isotropic Hardening Rule*

Continuous cyclic hardening has been observed for austenitic stainless steels for a large strain range under uniaxial loading [15]. To capture this behavior, we introduce the linear isotropic hardening rule:

$$dR = R_0(R_M)dp, \tag{17}$$

where parameter $R_0$ is dependent on the size of the memory surface:

$$R_0(R_M) = A_R R_M^2 + B R_M + C_R \text{ for } R_M \geq R_{M0}, \tag{18}$$

$$R_0(R_M) = A_R R_{M0}^2 + B R_{M0} + C_R \text{ otherwise,} \tag{19}$$

because of the strong dependence on the strain range observed in the experiments [15].

## 3. Identification of Material Parameters and Model Verification on SS304 Data

The cyclic plasticity model was implemented in the ANSYS FE code, using the algorithm described in [22]. The methodology for calibrating the proposed material model will be explained according to the classical Chaboche material model, which requires the following parameters to be identified: $\sigma_y$, $E$, $\mu$, $C_1$, $C_2$, $C_3$, $\gamma_1$, $\gamma_2$, and $\gamma_3$, where $E$ is the Young modulus and $\mu$ is the Poisson ratio.

Generally, 14 additional parameters have to be specified for the proposed cyclic plasticity model: $\phi_0$, $A_\infty$, $B_\infty$, $C_\infty$, $D_\infty$, $F_\infty$, $A_\omega$, $B_\omega$, $C_\omega$, $R_{M0}$, $R_{M\omega}$, $A_R$, $B_R$, $C_R$.

It is customary to determine the Young modulus $E$ from cyclic curves rather than from a tensile test. The Poisson ratio $\mu$ can be determined by standard procedures. The material parameter $E$ was established from the elastic region of the largest available hysteresis loop, using a linear regression. The initial yield strength $\sigma_y$ was chosen to get the best possible description of the static stress–strain curve using Equation (23).

A sequence of steps is retained that should be applied in the following description of the calibration of the proposed model. The sections are named according to the required experimental data. The parameters are identified on the basis of the experimental set of stainless steel SS304, available at [15].

*3.1. Uniaxial Large Hysteresis Loop*

It is well known that the material parameters of the Chaboche model can be determined (under cyclic loading) from the cyclic strain curve or from the large uniaxial hysteresis loop [23]. Figure 2 shows us the results for identifying the parameter in the case of stainless steel 304. According to [24], it is possible to use a relation that defines the loading part of the stabilized hysteresis loop in the stress—plastic strain diagram:

$$\sigma_x = \sigma_y + \alpha_{virt}, \tag{20}$$

$$\sigma_x = \sigma_y + \frac{C_1}{\gamma_1}\left(1 - 2e^{-\gamma_1(\epsilon_p - (-\epsilon_{pL}))}\right) + \frac{C_2}{\gamma_2}\left(1 - 2e^{-\gamma_2(\epsilon_p - (-\epsilon_{pL}))}\right) + C_3\epsilon_p, \tag{21}$$

where $\epsilon_{pl}$ is the plastic strain corresponding to the compressive peak strain and $\sigma_x$ is the axial stress. Relation (21) is valid in the case of $\gamma_3 = 0$ and for a large hysteresis loop.

If the Chaboche model is calibrated using the large hysteresis loop ($\Delta\epsilon = 6\%$), it predicts a higher stress amplitude for small strain amplitudes than was observed in the experiments. This phenomenon is shown for 1% strain amplitude in Figure 2. For this reason, a new cyclic plasticity model is needed. Parameters $C_i$ and $\gamma_i$ can be obtained for the newly-proposed model after applying Relation (21) to data from the largest available hysteresis loop, while parameter $\gamma_3 = 0$. In our case, we have used the Levenberg–Marquardt algorithm of the nonlinear least squares method. It is now clear that material parameters $C_i$ and $\gamma_i$ in the new model can be estimated on the basis of a single uniaxial hysteresis loop. The applicability of the new model to different strain ranges is given by the multiplier $\phi$ in Equation (11). Its evolution depends on memory surface size $R_M$ and accumulated plastic strain $p$. How the necessary parameters are identified will be explained below.

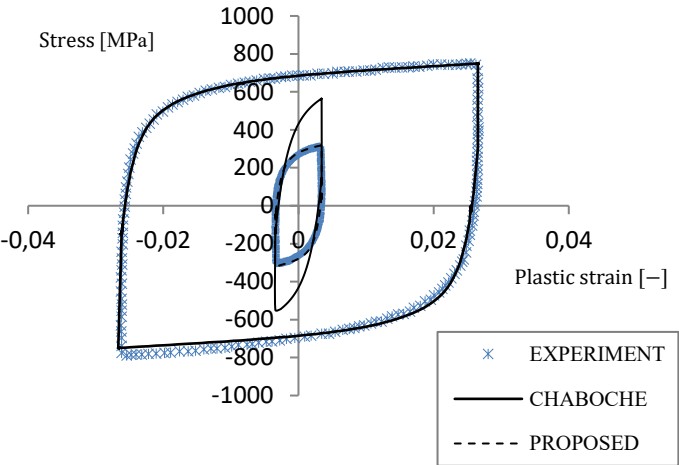

**Figure 2.** Prediction of two uniaxial hysteresis loops of SS304 (experimental data were taken from [15]).

### 3.2. Static Strain Curve

Under monotonic loading, the kinematic hardening rule of the proposed model is reduced to:

$$d\boldsymbol{\alpha}_i = \frac{2}{3}C_i d\boldsymbol{\epsilon}_p - \gamma_i \phi_0 \boldsymbol{\alpha}_i dp. \tag{22}$$

If isotropic hardening is neglected, the material parameter $\phi_0$ can be determined by a constitutive relation commonly used for the Chaboche model:

$$\sigma = \sigma_y + \frac{C_1}{\gamma_1 \phi_0}\left(1 - e^{-\gamma_1 \phi_0 \epsilon_p}\right) + \frac{C_2}{\gamma_2 \phi_0}\left(1 - e^{-\gamma_2 \phi_0 \epsilon_p}\right) + C_3 \epsilon_p. \tag{23}$$

For stainless steel SS304, the value of the parameter is $\phi_0 = 4.5$. The material model prediction of the static stress–strain curve is depicted in Figure 3.

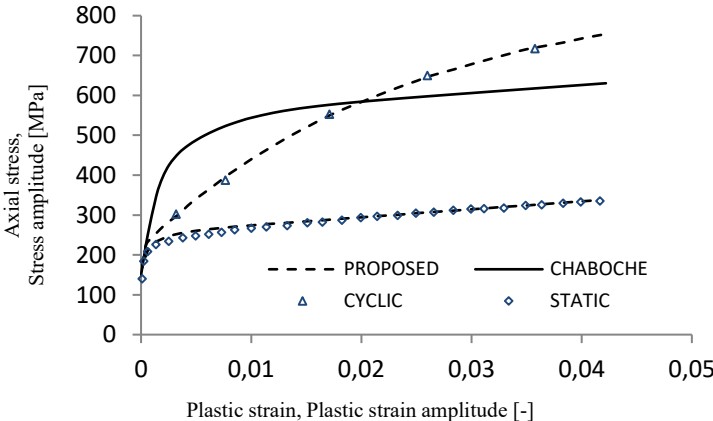

**Figure 3.** Prediction of static and cyclic uniaxial stress–strain curves; the experimental data for SS304 were reproduced from [15].

### 3.3. Cyclic Stress–Strain Curve

For the Chaboche model, the relation between stress amplitude and plastic strain amplitude can be derived, in accordance with [23], in the form:

$$\Delta\sigma/2 = \sigma_y + \frac{C_1}{\gamma_1}\tanh\left(\gamma_1\Delta\epsilon_p/2\right) + \frac{C_2}{\gamma_2}\tanh\left(\gamma_2\Delta\epsilon_p/2\right) + C_3\Delta\epsilon_p/2. \tag{24}$$

By analogy, the relation for the cyclic hardening curve can be obtained for the proposed constitutive model. Neglecting the isotropic part, we can write,

$$\Delta\sigma/2 = \sigma_y + \frac{C_1}{\gamma_1\phi_\infty}\tanh\left(\gamma_1\phi_\infty\Delta\epsilon_p/2\right)$$
$$+ \frac{C_2}{\gamma_2\phi_\infty}\tanh\left(\gamma_2\phi_\infty\Delta\epsilon_p/2\right) + C_3\Delta\epsilon_p/2. \tag{25}$$

This is a scalar nonlinear equation, which can be solved for selected experimental points, for example by the successive substitution method. Afterwards, the $\phi_\infty$ values for each peak of the hysteresis loop are fitted by the approximate function (13).

The experimental cyclic stress–strain curve data considered without linear isotropic hardening (published in [15]) and the predicted data corresponding to the proposed model are also shown in Figure 3. The cyclic stress–strain curve of the classic Chaboche model, calibrated using the large hysteresis loop ($\Delta\epsilon = 6\%$), is also presented in Figure 3. It is again clear that a more robust model is needed.

### 3.4. Cyclic Hardening Curves

In order to provide a good description of the cyclic hardening properties for a wide range of strain amplitudes, it is necessary to identify the isotropic hardening and kinematic hardening functions.

The remaining parameters $A_\omega$, $B_\omega$, $C_\omega$, $R_{M\omega}$, $R_{M0}$, $A_R$, $B_R$, and $C_R$ were estimated by a fitting procedure, using the nonlinear relations between the peak stress and the accumulated plastic strain $p$ for all available cases of constant strain amplitude tests for the particular type of stainless steel ($\Delta\epsilon = 1, 2, 4, 6$, and $8\%$).

The isotropic hardening parameters were determined from the slope of each cyclic hardening/softening curve in a saturated state. More precisely, a unique value for each case of a strain range was obtained, which was afterwards used for approximation by (18) and (19). The estimated material parameters of the proposed cyclic plasticity model are stated in Table 1.

Note that parameter $\gamma_3$ was equal to 10, which corresponds more to the behavior of metallic materials, e.g., if ratcheting occurs under stress-controlled loading with a nonzero mean axial stress value.

**Table 1.** Material parameters of the proposed model for SS304.

| $E$ [MPa] | $\nu$ | $\sigma_y$ [MPa] | $C_1$ [MPa] | $\gamma_1$ | $C_2$ [MPa] |
|---|---|---|---|---|---|
| 196,000 | 0.3 | 150 | 150,000 | 622 | 19,827 |
| $\gamma_2$ | $C_3$ [MPa] | $\gamma_3$ | $A_\infty$ | $B_\infty$ | $C_\infty$ |
| 128 | 2000 | 10 | 0 | $1.15 \times 10^{-7}$ | $-1.23 \times 10^{-4}$ |
| $D_\infty$ | $F_\infty$ | $A_R$ [MPa$^{-1}$] | $B_R$ | $C_R$ [MPa] | $R_{M0}$ [MPa] |
| 0.032 | $-3.6$ | 0.000915 | $-0.5$ | 60.7 | 305 |
| $A_\omega$ | $B_\omega$ | $C_\omega$ | $R_{M\omega}$ [MPa] | $\phi_0$ | |
| 0 | $4.02 \times 10^{-17}$ | 6.424 | 344 | 4.5 | |

### 3.5. Prediction Results for SS304

For implementation in ANSYS, the user subroutine called USERMAT1D.F, which was originally distributed for bilinear isotropic hardening. It was necessary to modify the user subroutine according to the used radial return algorithm [22]. A single LINK180 element was used for all analyses in ANSYS, because only uniaxial loading cases are considered in this paper. Material SS304, which was used as an example to explain the calibration procedure, exhibited very strong cyclic hardening at larger amplitudes of plastic strain. All investigated cases corresponded to the uniaxial experiments published by Kang et al. [15]. Predictions of cyclic hardening, corresponding to the proposed model, are shown together with the results of experiments in the form of peak tensile stress values as a function of the number of cycles (Figure 4).

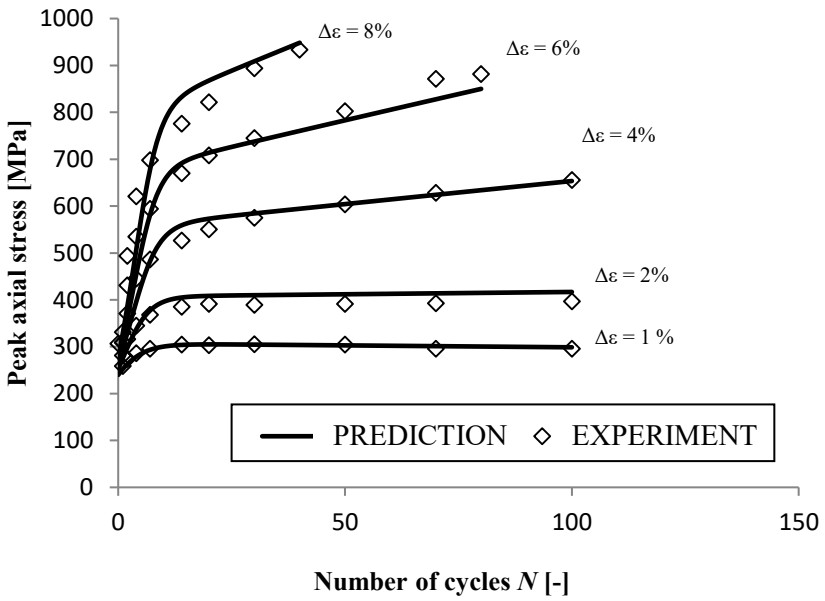

**Figure 4.** A comparison of simulations and experiments in the form of tensile peak stress variation; uniaxial strain controlled tests (the experiment was taken from [15]).

The results of the prediction of the transition behavior of the SS304 steel material during uniaxial cyclic loading (Figure 4) were supplemented by hysteresis loops for strain amplitudes of 1% and 6%, respectively (Figures 5 and 6). These results can be compared with the experimentally-obtained hysteresis loops published in [15].

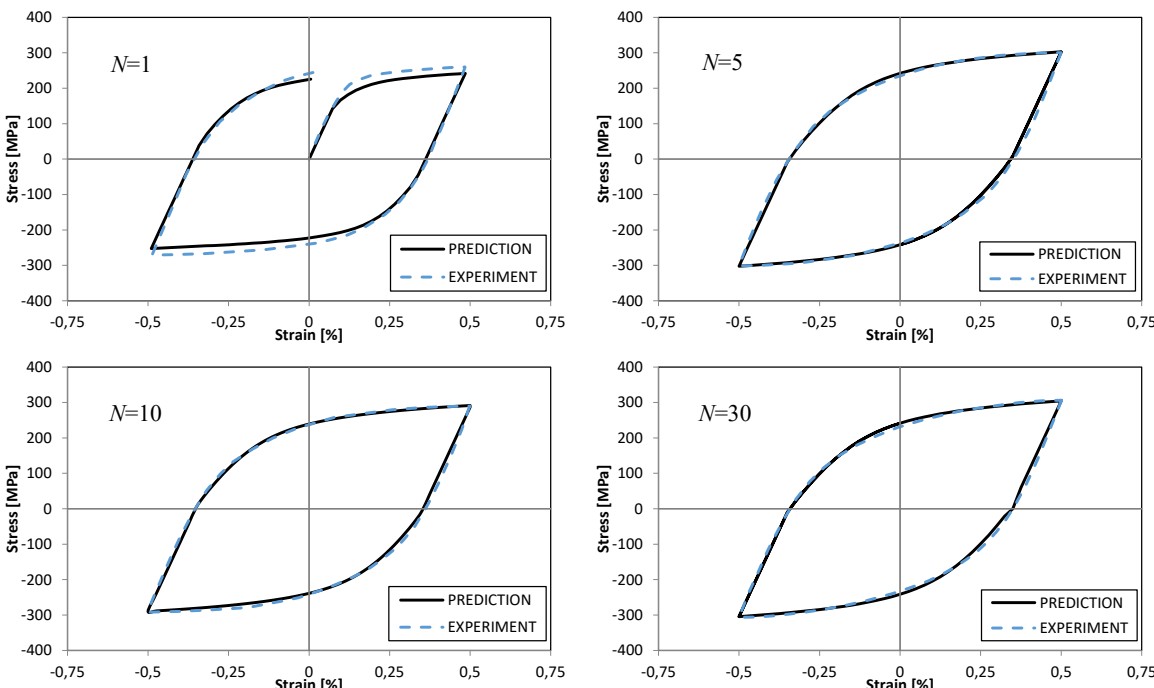

**Figure 5.** Prediction of the uniaxial test with $\Delta\epsilon = 1\%$ (experimental data were taken from [15]).

Another simulation was of a cyclic test with a linearly-increased/-decreased strain amplitude composed of five identical blocks. In each block, the strain range was increased within 20 cycles to a value of 5%, and subsequently, it was reduced, with the same increment. The resulting stress–total strain dependence for the strain range increasing stage is shown in Figure 7. The increasing amplitude of the stress was more progressive in the prediction than in the experiment, as can also be seen in Figure 8.

Figure 8 presents a comparison with an experiment, in which the peaks of the hysteresis loops are plotted for the first and fifth loading block. The relative error of the prediction was about 12.3% in the first block (for amplitude of strain 2.5%), but it was reduced to 6.7% in the fifth loading block.

The proposed model was able to capture the static and cyclic stress–strain curve for SS304 correctly. It also simulated well the shapes of the stress–strain hysteresis loops in all investigated cases, as well as the Bauschniger effect, which became weaker for higher strain ranges. The non-Masing behavior of the SS304 material was very strong, and this can be modeled better with superposition of the kinematic hardening and isotropic hardening, as in the proposed constitutive model. For the incremental test, the prediction was very good, especially in the fifth loading block. The overprediction of the peak stresses in the first block of loading can be reduced, for example, by introducing memory surface contraction, as was proposed in the original model of Jiang and Sehitoglu [13].

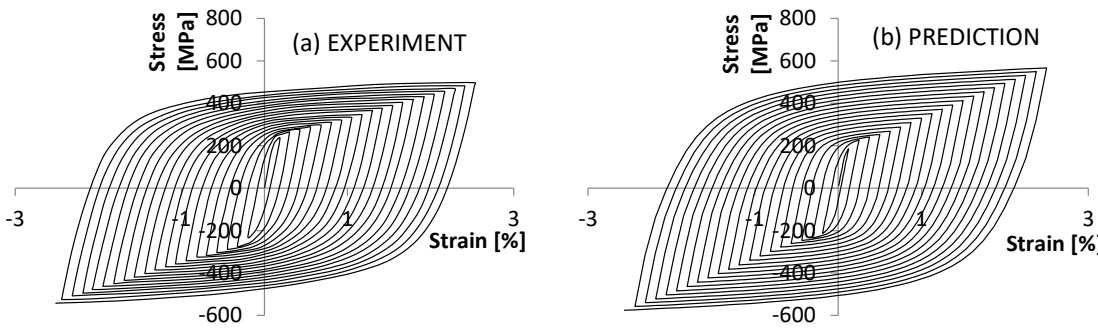

**Figure 6.** Prediction of the uniaxial test with $\Delta\epsilon = 6\%$ (experimental data were taken from [15]).

**Figure 7.** Prediction of the incremental test with the linearly-increasing amplitude of the axial stress: (**a**) experiment (experimental data were taken from [15]); (**b**) prediction.

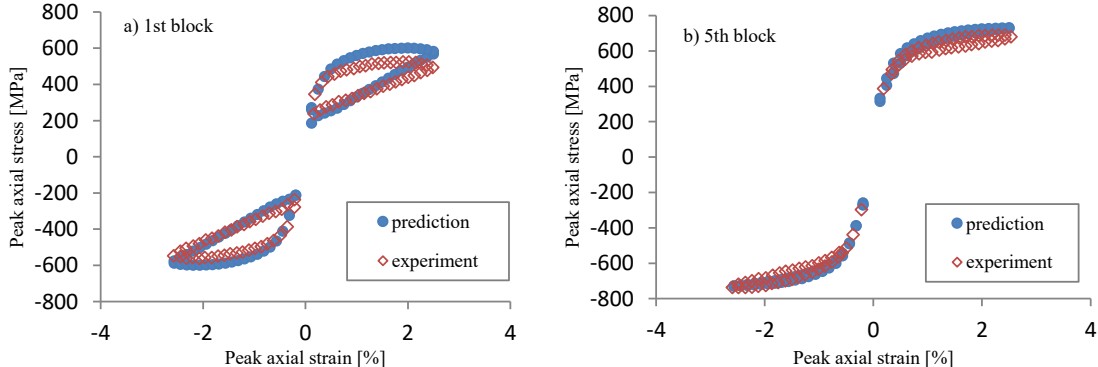

**Figure 8.** Comparison between a prediction and an experiment in the form of the variation in tensile peak stress for the incremental test (the experiment was taken from [15]): (**a**) first block, (**b**) fifth block.

## 4. Application to Uniaxial Cyclic Tests of 08Ch18N10T Stainless Steel

### 4.1. Identification of the Material Parameters for 08Ch18N10T Stainless Steel

The model was also calibrated for original experimental data on austenitic steel 08Ch18N10T [25]. A total of 12 uniaxial specimens were used for the material parameters' identification process (marked by the abbreviation IDF in the following text, each specimen representing a different level of loading). According to the ASTM standard [26], the classic uniform-gauge geometry of the specimen is limited up to the amplitude of the total strain $\epsilon_a = 0.5\%$. For higher strain levels, an hour-glass type geometry is required. According to this standard, the IDF specimens were compiled from uniform-gauge geometry (specimens IDF1–IDF5) and hourglass geometry (specimens IDF6–IDF12); see Figure 9.

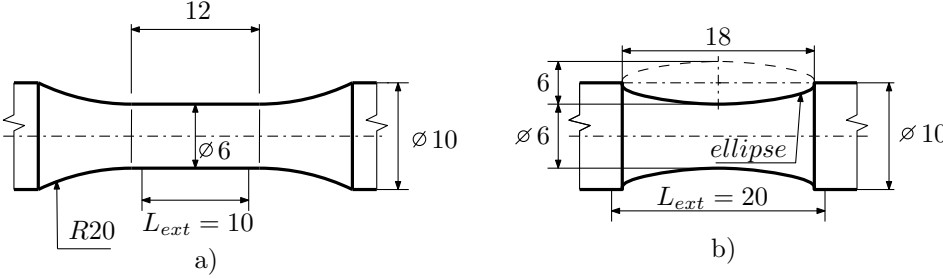

**Figure 9.** Specimen geometries: (**a**) uniform-gauge; (**b**) hourglass.

Another 17 uniaxial specimens (all with hourglass geometry) were used to verify the prediction ability of the model.

The loading force *F* applied to the IDF specimen was known, as was the strain field of the surface of the specimen. The strain field was measured by the extensometer in the case of uniform-gauge geometry, or by the digital image correlation method in the case of hourglass geometry. Considering the uniaxial stress field in the cross-section of a specimen, the stress can be determined as $\sigma = F/A$, where *A* is the cross-section surface of the specimen. This allowed the use of a different calibration process, based on knowledge of the shape of the stress–strain hysteresis loops in all cycles during the experiment to failure.

Let us select one hysteresis stress–strain loop of a point on the specimen representing one loading cycle. This can be optimally simulated by a set of material parameters $C_1$, $\gamma_1$, $C_2$, $\gamma_2$, $C_3$, $\gamma_3$, and $\sigma_y$. However, in the next cycle, the optimal set of these parameters can be slightly different, as can the set of parameters of a specimen with different loading conditions. This material model uses the memory surface concept by setting these material parameters as functions of $R_M$ and making these coefficients dependent on the loading history and the loading level conditions.

The material model did not include a simulation of the material damage process, so only experimental data up to damage were used for the calibration. The number of cycles used was $N_d$, and this number corresponded to the drop in the loading force during the experiment by 2%, due to crack initiation and propagation, leading to failure.

First, the fatigue life was divided into about 10 evenly-spaced parts by selecting hysteresis loops (SHLs), and the cycle number of each selected hysteresis loop (SHL) is given as $n \simeq N_d/k$, where $k = 1, 2, \ldots, 10$. The Young modulus $E$, the Poisson ratio $\nu$, and the yield strength $\sigma_y$ were determined from the tensile test according to the ISO standard [27].

$\sigma_y$ can be interpreted as the point where the linear part of the tensile curve turns into the non-linear part (see Figure 10). The root mean squared error method ($RMSE$) can be applied to find the point. In the tensile test (or in the first cycle of the cyclic test), the yield strength $\sigma_y$ corresponded to $RMSE \approx 8$. Applying $RMSE = 8$ to each SHL, the actual yield stress $Y$ was found. This shows the development of the actual yield stress $Y$ during the fatigue life; see Figure 11.

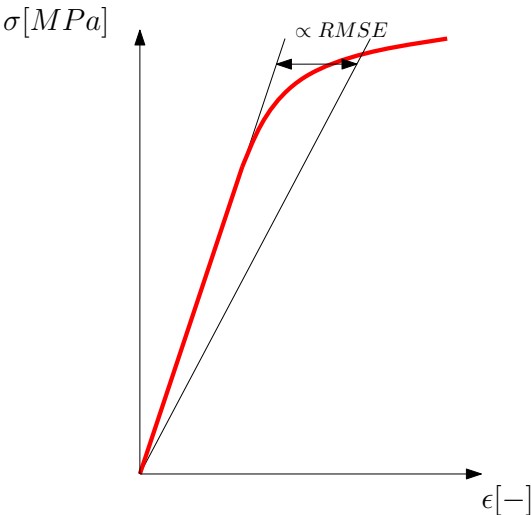

**Figure 10.** Actual yield stress determination.

Two SHLs were chosen, the bigger one and the smaller one, each with cycle number $n = N_d$ (the last cycle). The Chaboche material model parameters $C_1$, $\gamma_1$, $C_2$, $\gamma_2$, $C_3$, and $\gamma_3$ were found using an optimization process. The target function was set to the optimal shape match between the simulation and experiment of the two SHLs. The result is shown in Figure 12.

Knowing the Chaboche material parameters, a first guess of the memory surface size for each specimen was determined, using Equations (5)–(9). The formulation of $R_M$ and the constant amplitude of the loading conditions resulted in fast saturation of the $R_M$ value for each specimen (after the first cycle), which made the calibration process easier.

The yield stress was now fitted as a function of $R_M$, using Equations (17)–(19), by finding material parameters $A_R$, $B_R$, and $C_R$. Using the tensile test experimental data and performing a simulation of this test, parameter $\phi_0$ was found using Equation (11) as an optimal value of $\phi$ for the tensile test simulation. The value of function $\phi$ from Equation (11) was found for SHLs, using a similar optimization process as for determining the Chaboche material parameters. $\phi_\infty$ was the value of $\phi$ for $n = N_d$, and from Equation (14), $\phi_\infty$ was then set as a function of $R_M$ by finding material parameters $A_\infty$, $B_\infty$, $C_\infty$, $D_\infty$, and $F_\infty$. Function $\omega$ determined the transition of function $\phi$ between its border values $\phi_0$ and $\phi_\infty$. Knowing the course of function $\phi$ during the fatigue life, $\omega$ was determined as a function of $R_M$ by finding material parameters $A_\omega$, $B_\omega$, and $C_\omega$ from Equation (15). This result was not necessarily optimal, so one more optimization was performed to find better $\phi_\infty$ and $\omega$ material parameters. The target function was set to the best possible match of the amplitude stress response between simulation and experiment during the whole fatigue life (not only SHLs).

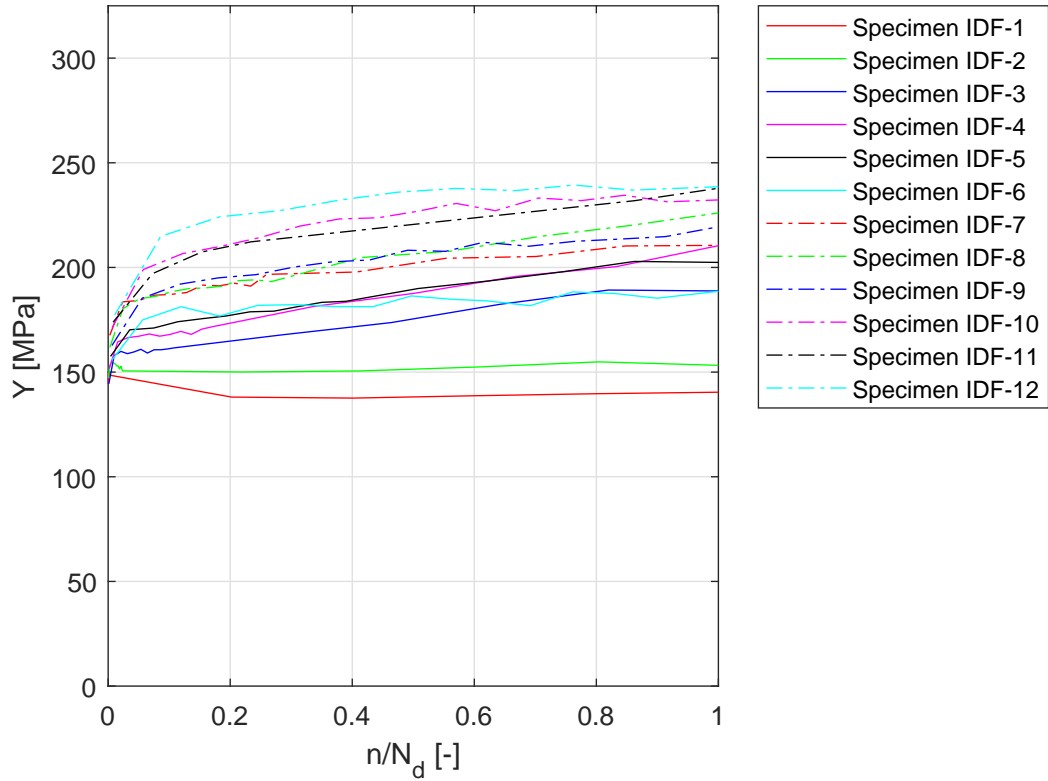

**Figure 11.** Actual yield stress development during fatigue life evaluated for 08Ch18N10T.

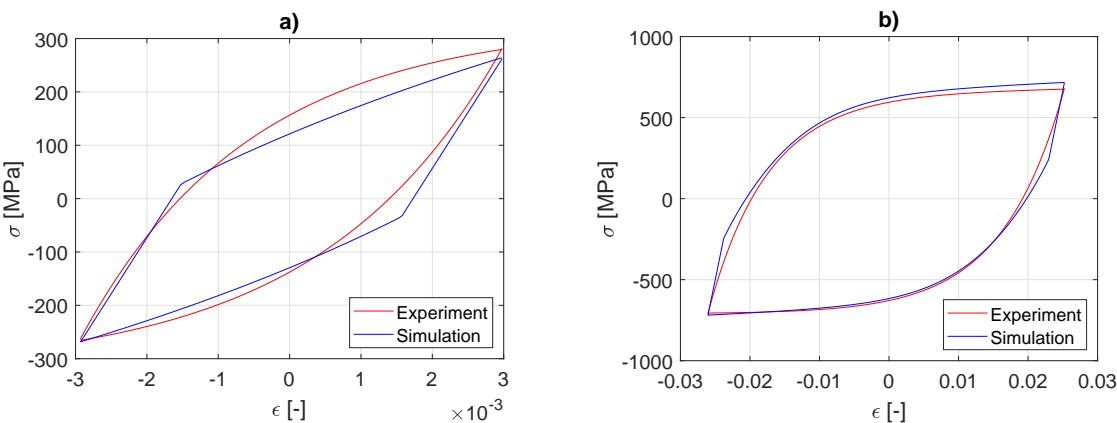

**Figure 12.** Chaboche coefficients fitting: (**a**) small loop; (**b**) large loop.

The $R_M$ value for each specimen was determined only as a first guess, so a number of iterations of the whole calibration process had to be carried out to find the final and optimal set of material parameters. The optimal material parameters are presented in Table 2.

**Table 2.** Material parameters of the proposed model for 08Ch18N10T.

| $E$ [MPa] | $\nu$ | $\sigma_y$ [MPa] | $C_1$ [MPa] | $\gamma_1$ | $C_2$ [MPa] |
|---|---|---|---|---|---|
| 210,000 | 0.3 | 150 | 63,400 | 148.6 | 10,000 |
| $\gamma_2$ | $C_3$ [MPa] | $\gamma_3$ | $A_\infty$ | $B_\infty$ | $C_\infty$ |
| 911.4 | 2000 | 0 | $-1.441 \times 10^{-9}$ | $1.911 \times 10^{-6}$ | $-8.951 \times 10^{-4}$ |
| $D_\infty$ | $F_\infty$ | $A_R$ [MPa$^{-1}$] | $B_R$ | $C_R$ [MPa] | $R_{M0}$ [MPa] |
| $1.688 \times 10^{-1}$ | $-10.6$ | $1.264 \times 10^{-4}$ | $-4.709 \times 10^{-2}$ | 3.801 | 225.4 |
| $A_\omega$ | $B_\omega$ | $C_\omega$ | $R_{M\omega}$ [MPa] | $\phi_0$ | |
| 0 | $3.456 \times 10^{-11}$ | $-4.197$ | 130.5 | 2.318 | |

### 4.2. Uniaxial Prediction for 08Ch18N10T Stainless Steel

The proposed model was used for an FE simulation of the uniaxial experimental program. The error of the amplitude of the force between experiment and simulation is formulated as:

$$Error = \frac{F_{a\ exp} - F_{a\ sim}}{F_{a\ exp}} \cdot 100 [\%] \tag{26}$$

The mean error over the specimen is defined as:

$$Mean\ Error = \frac{1}{N_d} \sum_{n=1}^{N} Error_n \tag{27}$$

where $Error_n$ is the error in cycle $n$ and $N$ is the overall number of cycles in the simulation. The total error over all specimens is calculated as:

$$Total\ Error = \frac{1}{S} \sum_{s=1}^{S} Mean\ Error_s = 4.83\% \tag{28}$$

where $Mean\ Error_s$ is the mean error of specimen number $s$ and $S = 17$ is the total number of specimens. The results of the experiments and the FEA simulations of all 17 uniaxial specimens are shown in Figures 13–29.

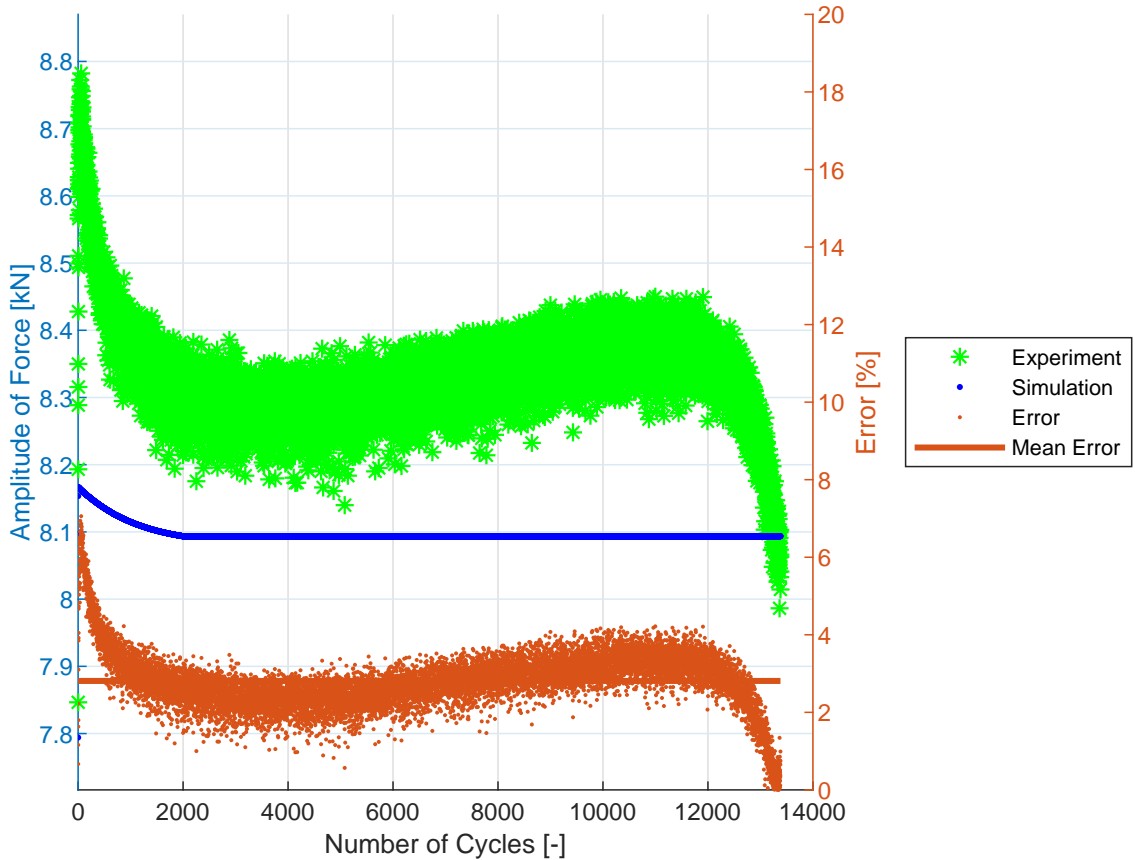

**Figure 13.** Specimen E9-1.

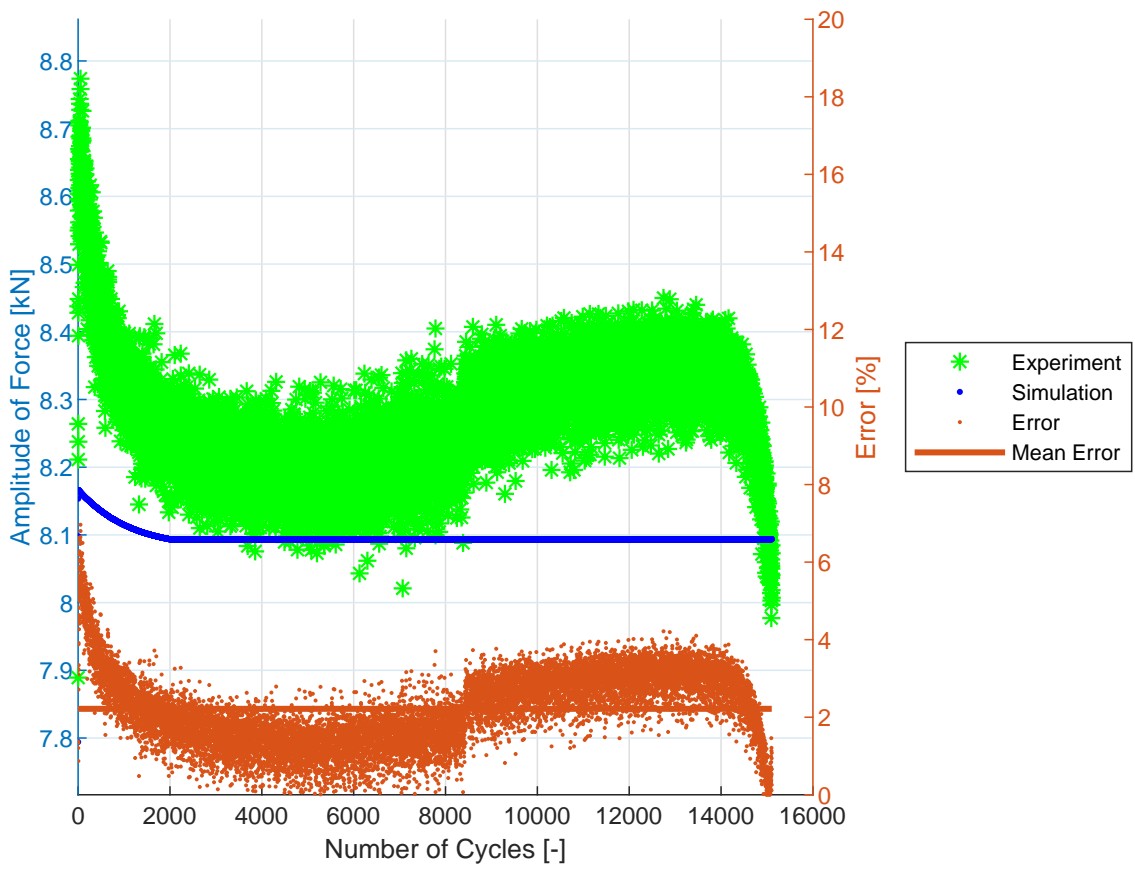

**Figure 14.** Specimen E9-2.

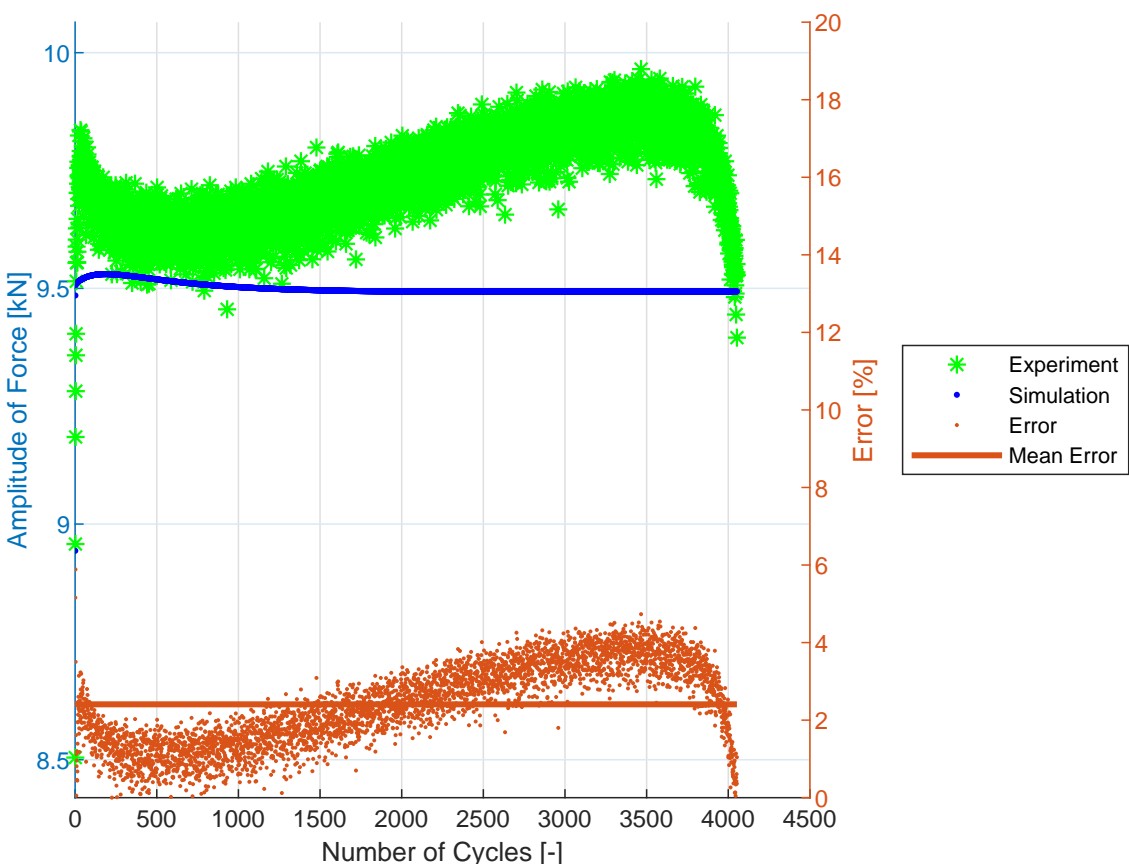

**Figure 15.** Specimen E9-3.

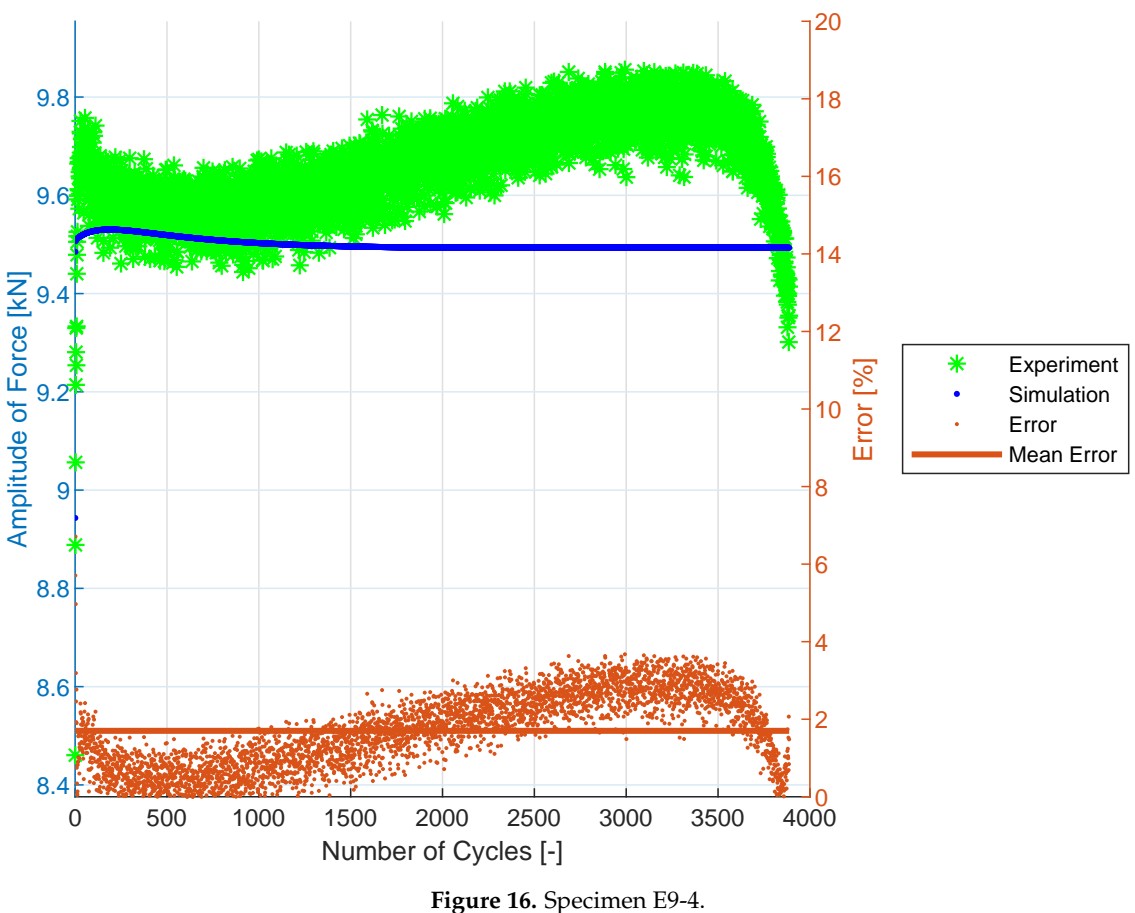

**Figure 16.** Specimen E9-4.

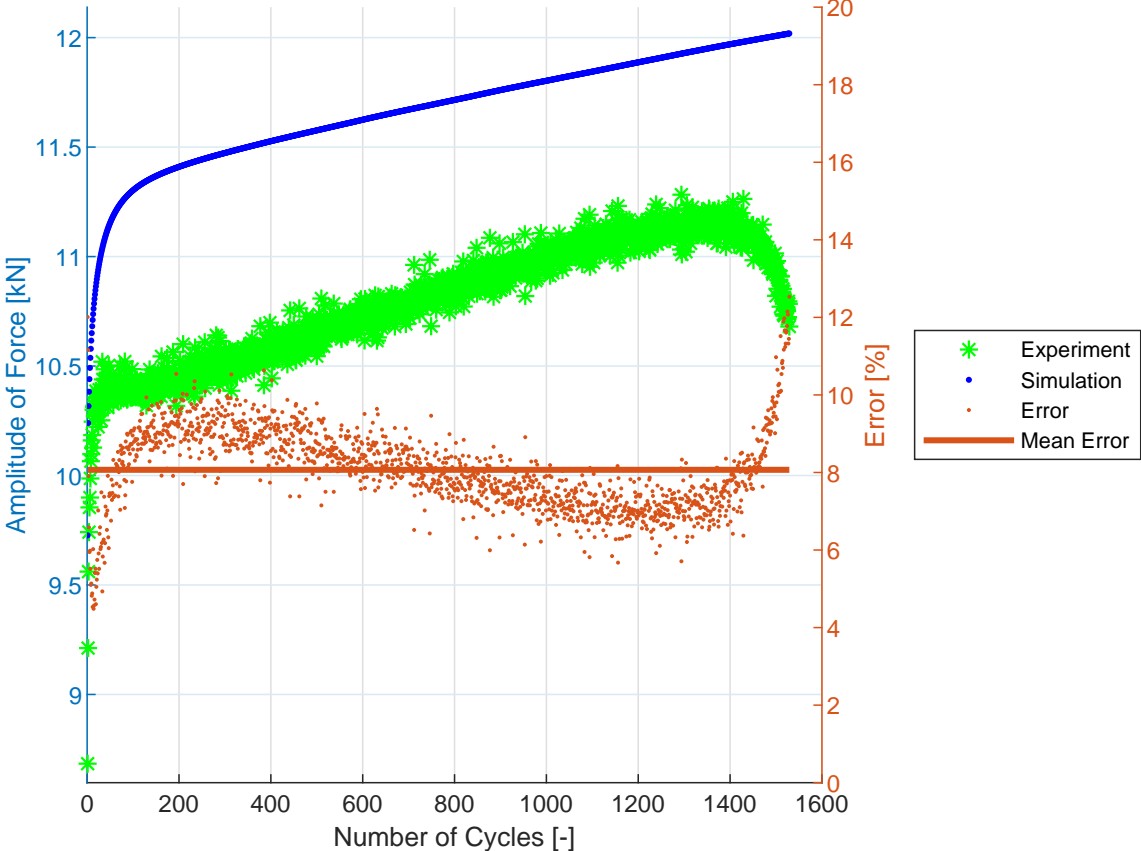

**Figure 17.** Specimen E9-5.

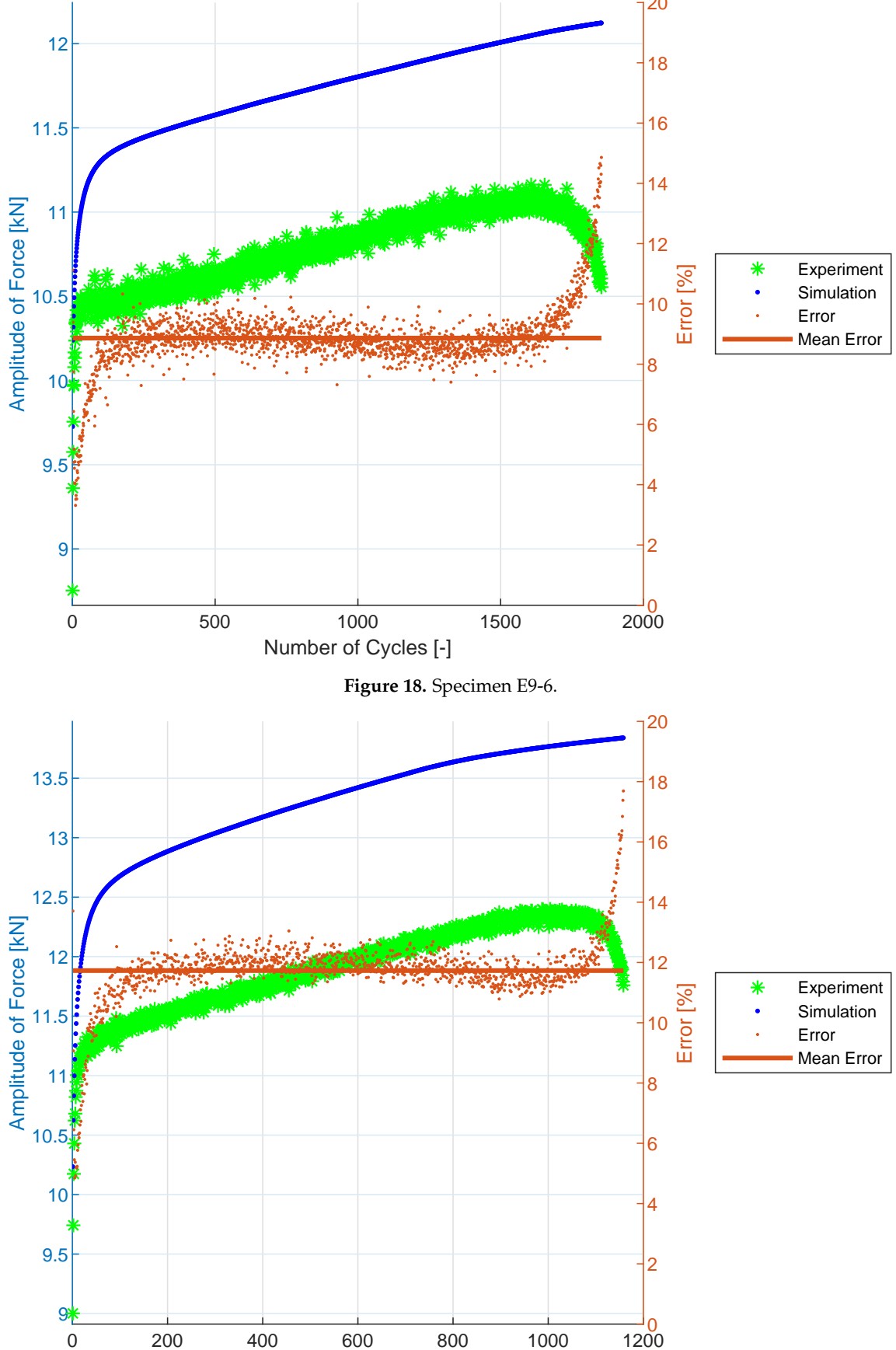

**Figure 18.** Specimen E9-6.

**Figure 19.** Specimen E9-7.

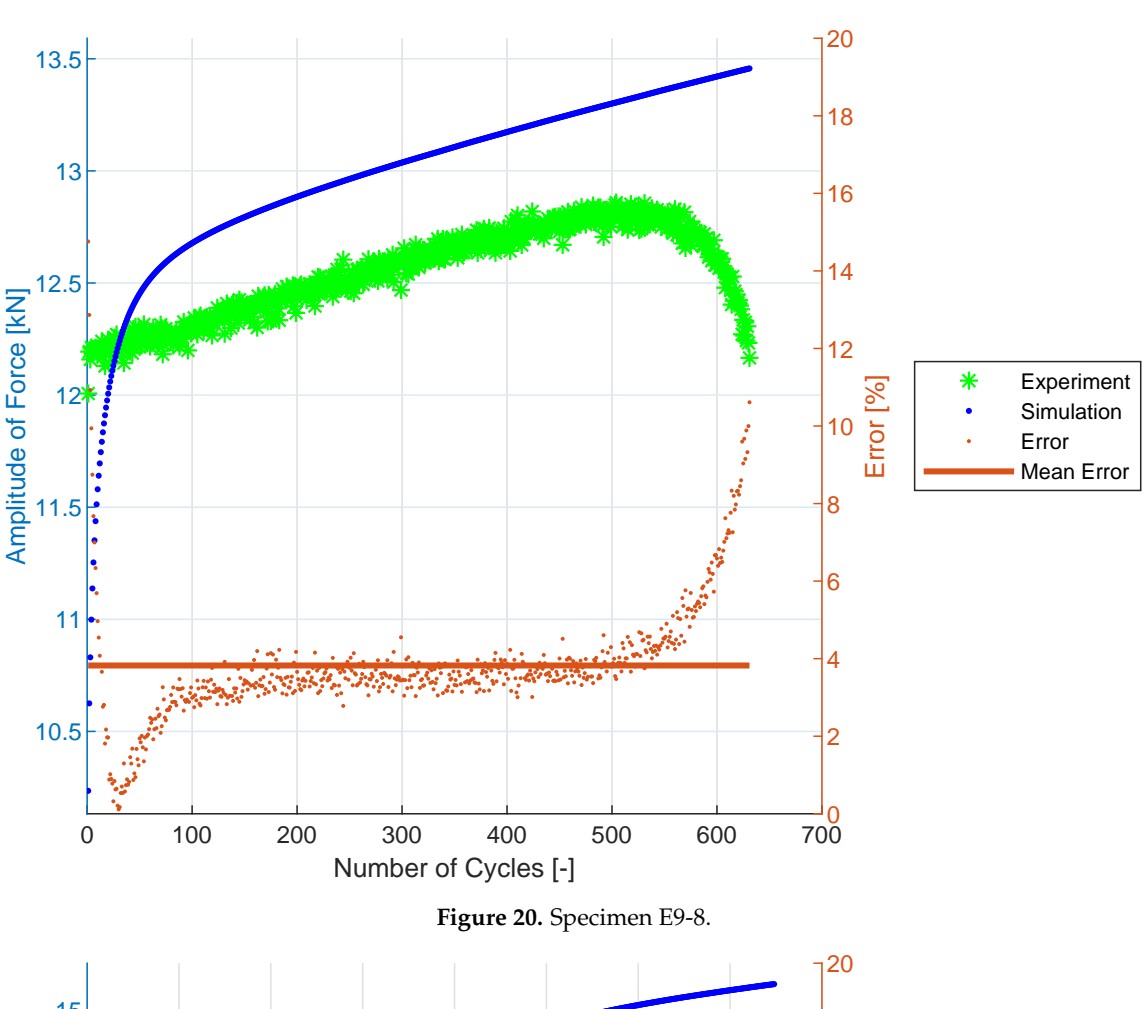

**Figure 20.** Specimen E9-8.

**Figure 21.** Specimen E9-9.

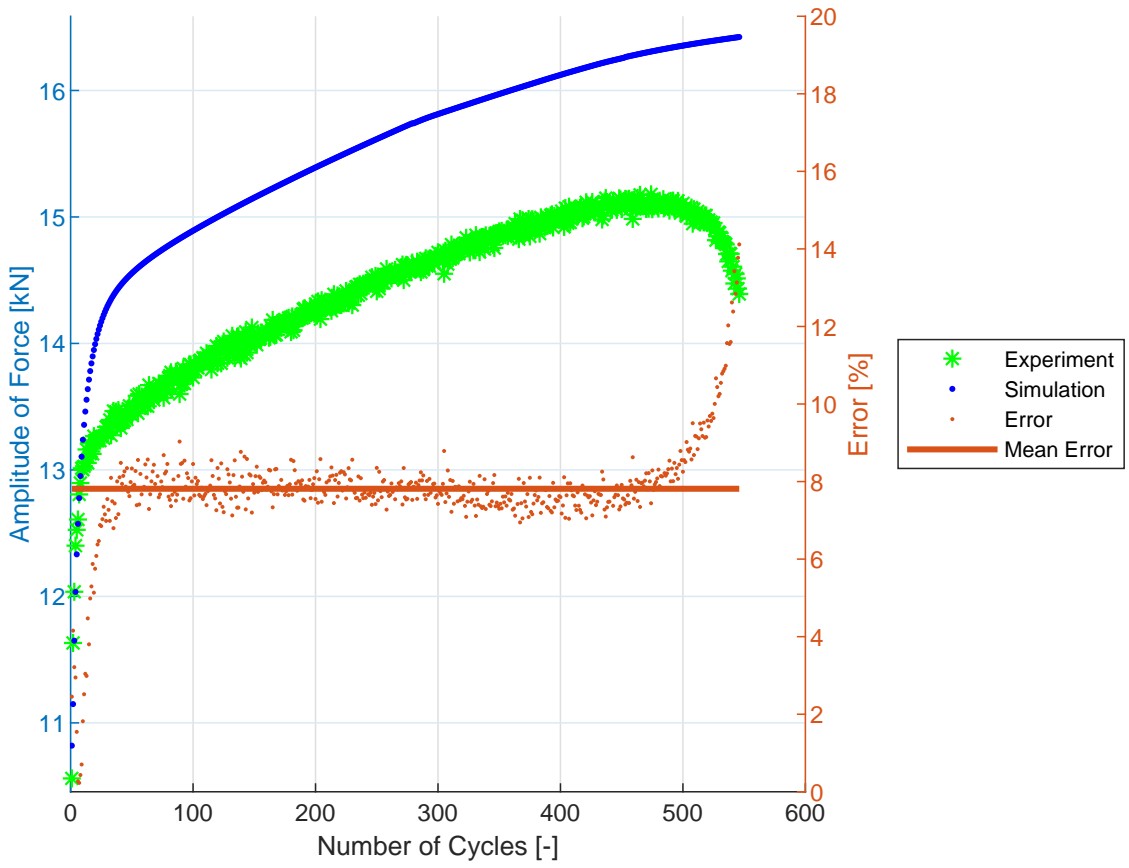

**Figure 22.** Specimen E9-10.

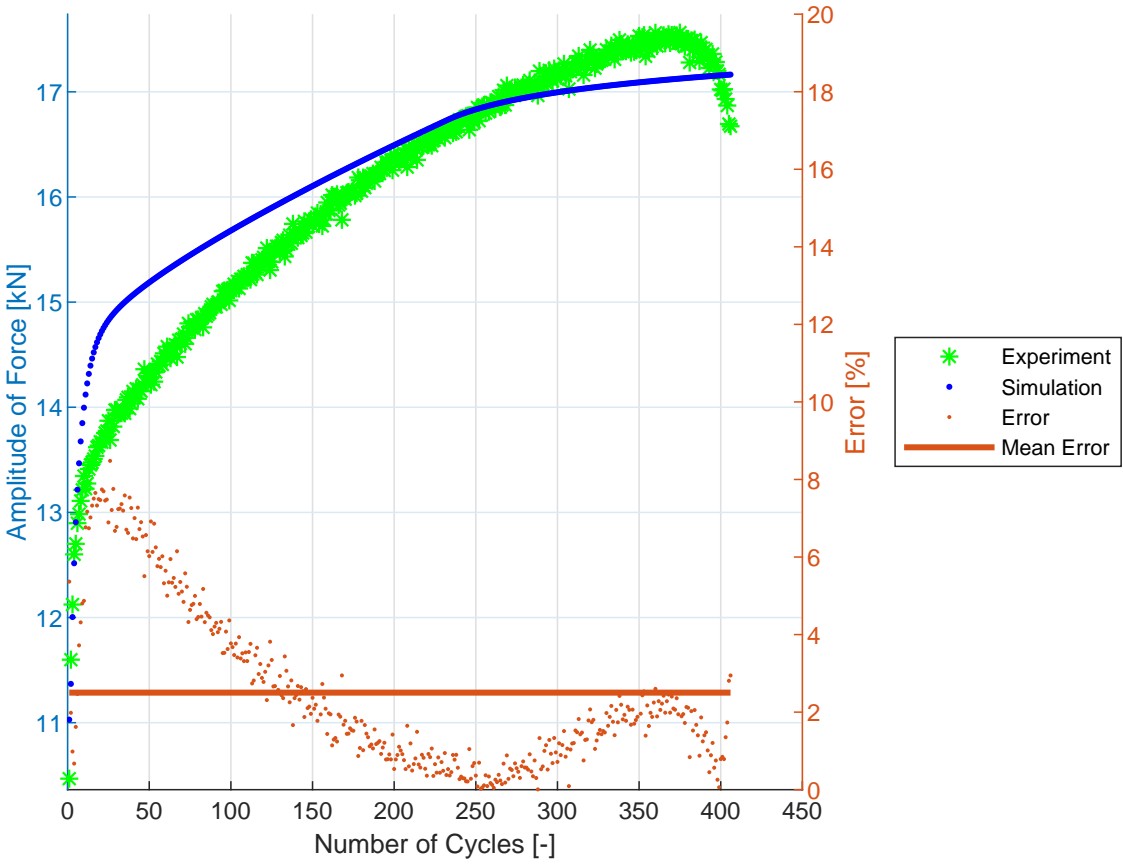

**Figure 23.** Specimen E9-11.

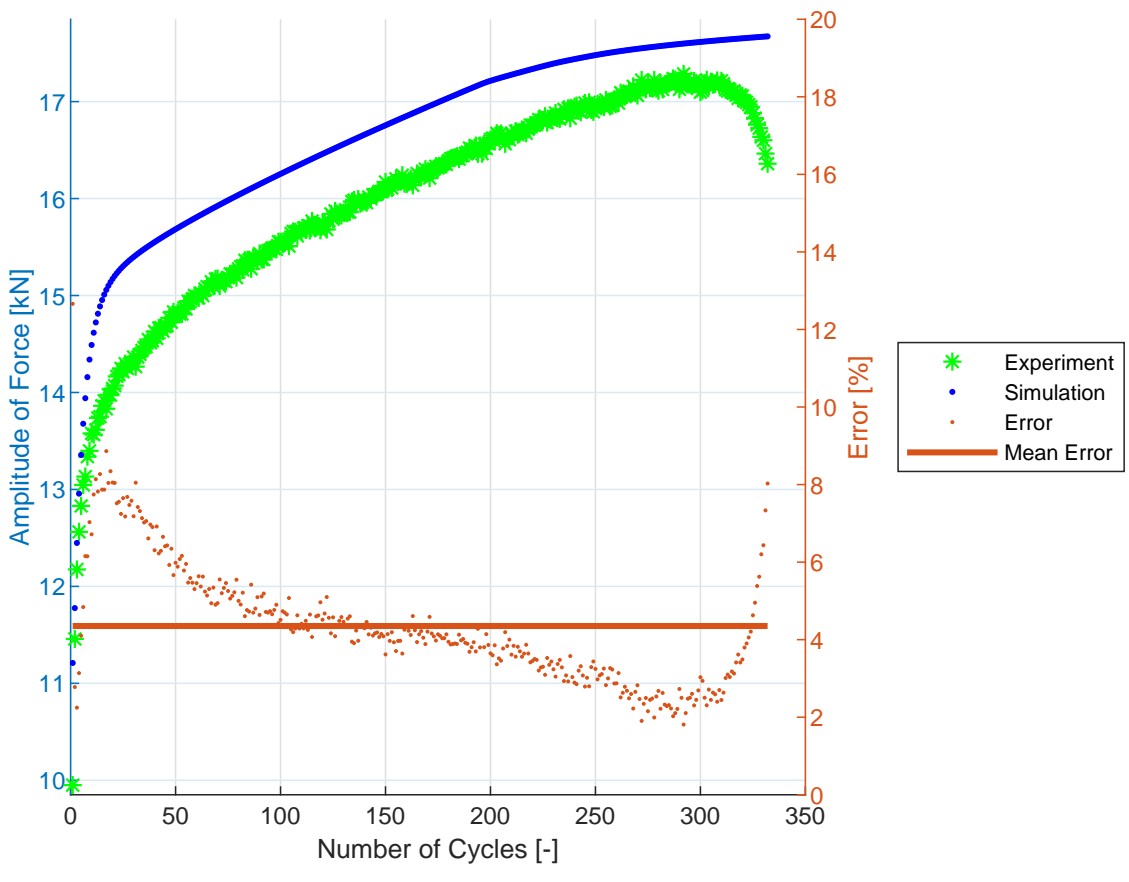

**Figure 24.** Specimen E9-12.

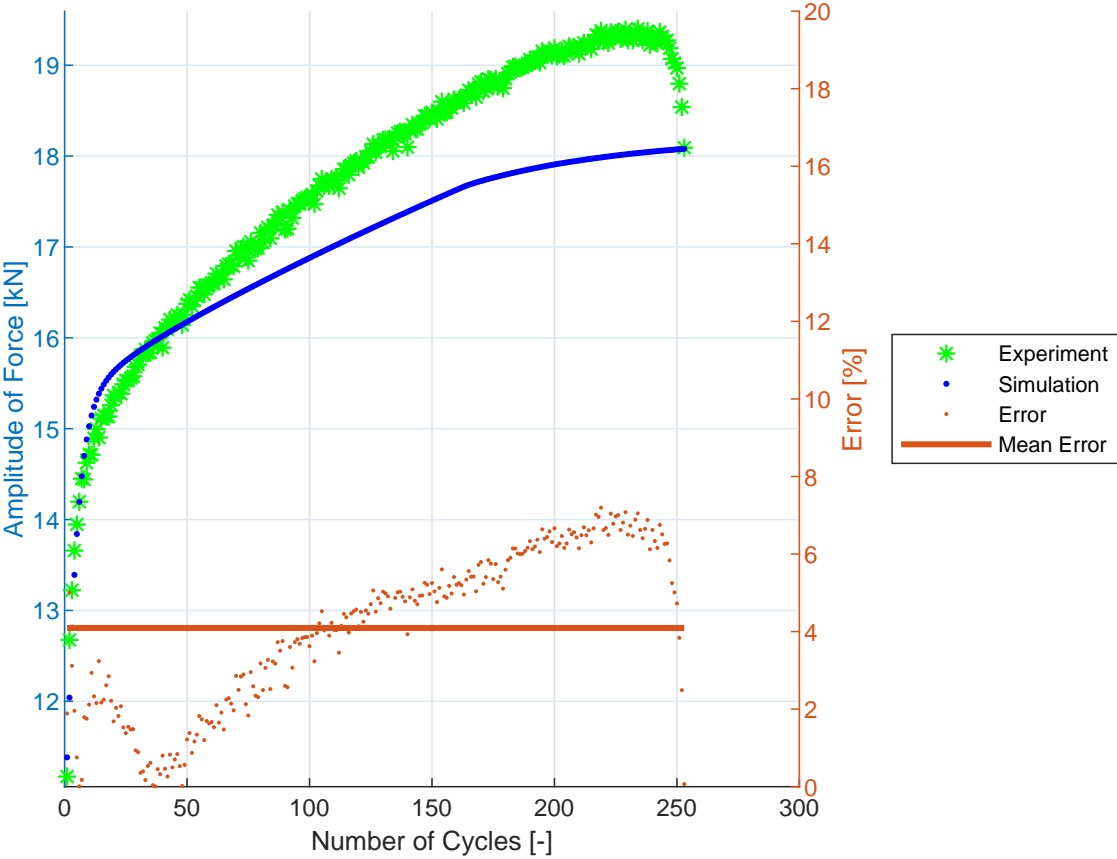

**Figure 25.** Specimen E9-13.

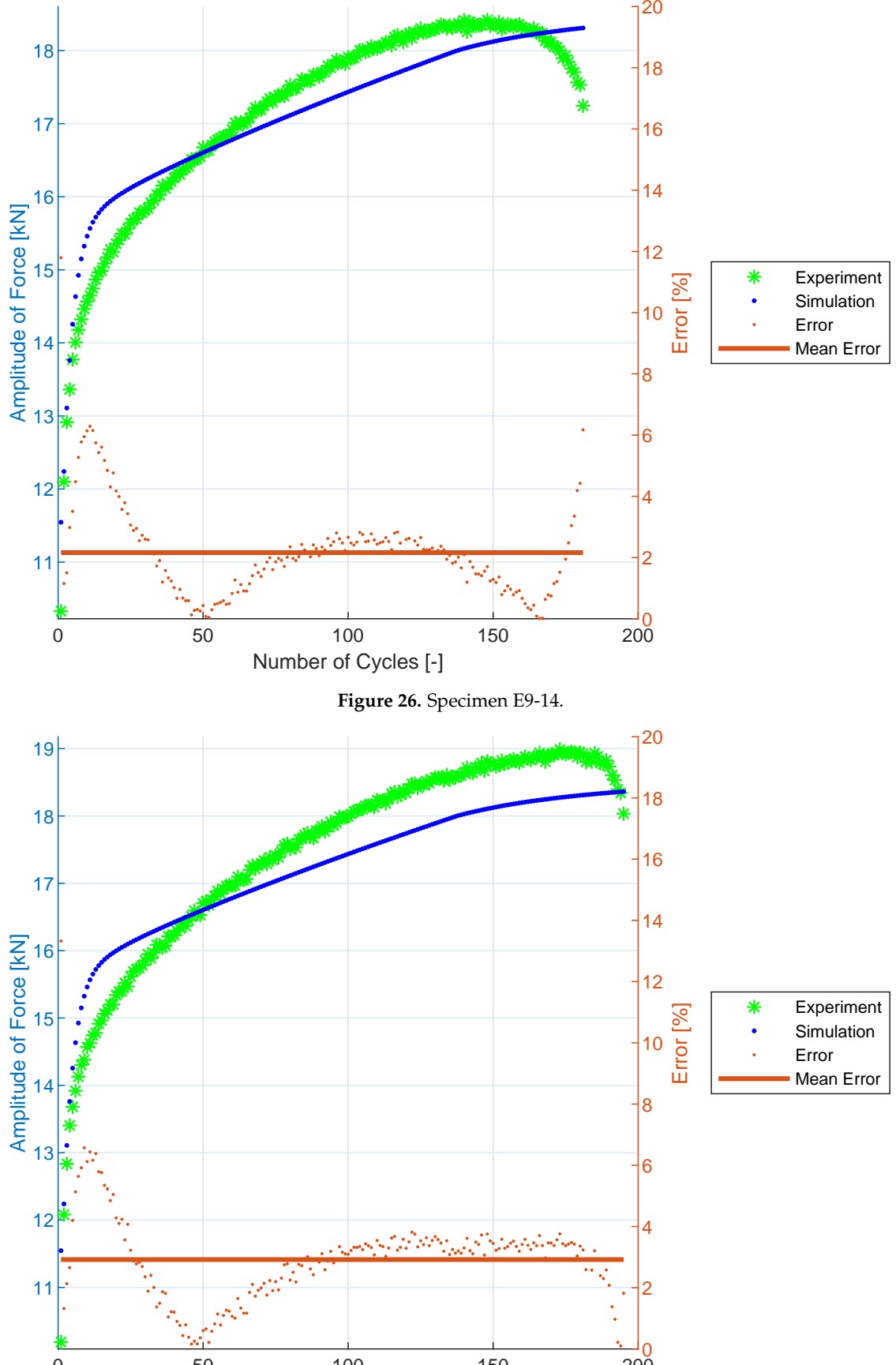

**Figure 26.** Specimen E9-14.

**Figure 27.** Specimen E9-15.

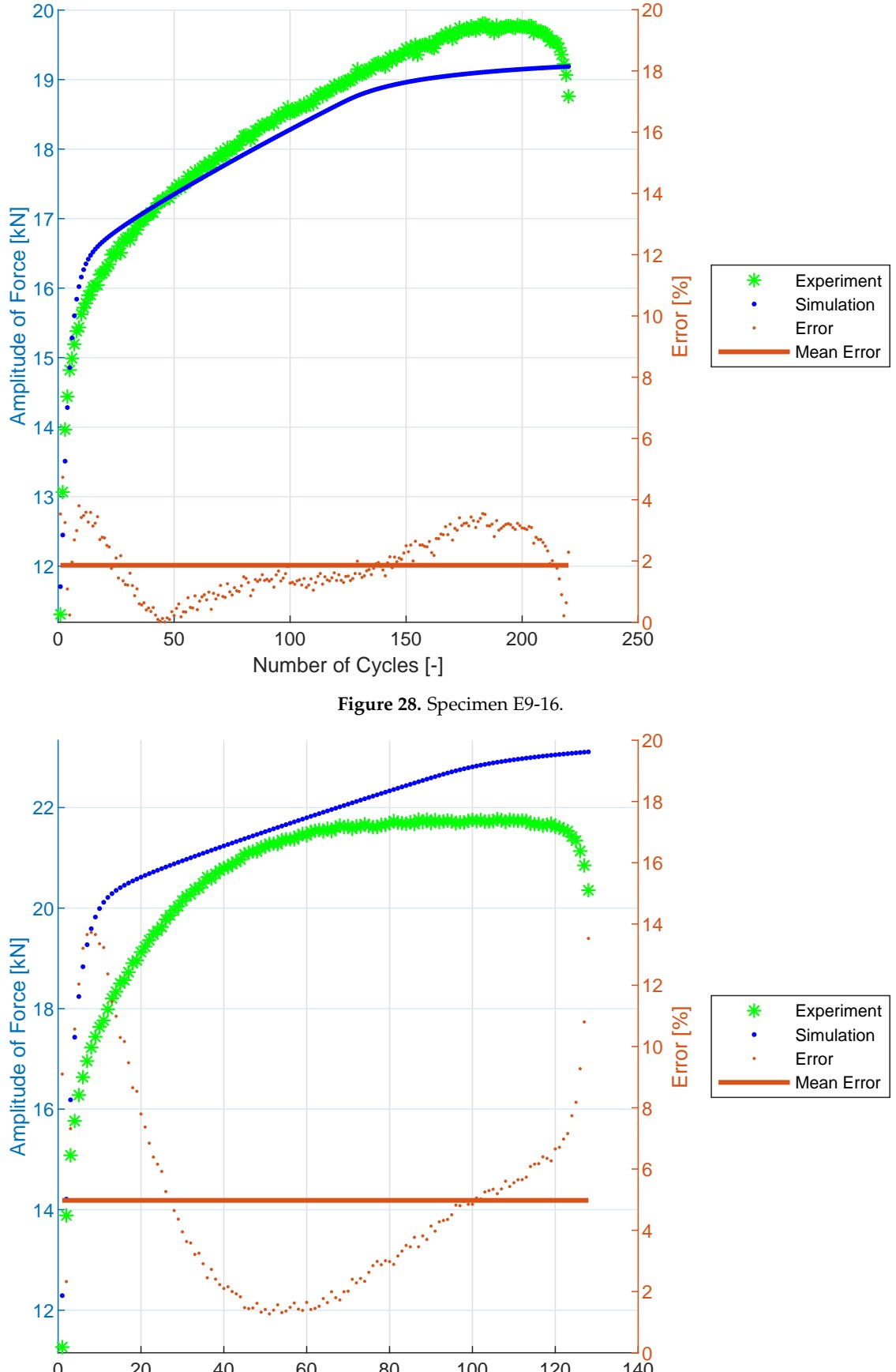

**Figure 28.** Specimen E9-16.

**Figure 29.** Specimen E9-17.

## 5. Discussion

As was shown in the previous section, the proposed model was able to capture the static curve and the cyclic stress–strain curve of SS304 very well. It also simulated well the shapes of the stress–strain hysteresis loops in all investigated cases, as well as the Bauschniger effect. The non-Masing behavior of the SS304 material was very strong, and this can be modeled better by superposing the kinematic and isotropic hardening, as proposed in the new constitutive model.

For the incremental test, the prediction was very good, especially in the fifth loading block, where it outperformed Kang's model [15]. Overprediction of the peak stresses in the first block of loading can be reduced, e.g., by introducing the memory surface contraction, as was proposed in the original model of Jiang and Sehitoglu [13].

The proposed model also provided a good description of the uniaxial tests of austenitic steel 08Ch18N10T. It captured the strain range-dependent cyclic hardening of this material, with an average simulation error of 4.83%. The proposed model slightly overestimated the initial phase of hardening. Furthermore, the model was unable to describe the softening at the end of the fatigue life caused by the fatigue crack growth. This phenomenon was not included in the model.

Figure 14 (Specimen E9-2) shows the attentive reader what seems to be a jump on the error axis at about 8200 cycles. Zooming on the data shows that the amplitude of the force predicted by the simulation was constant, while there was a relatively small gradual increase in the amplitude of the force in the experiment that took place over dozens of cycles. This is a common phenomenon in cyclic testing. Along with the relatively small value of the error between experimental prediction and simulation in that area, it optically intensified the jump effect in the graph.

## 6. Conclusions

In this paper, a new model of cyclic plasticity was proposed for describing the cyclic hardening of a material, when there is an influence of strain amplitude, based on the Jiang–Sehitoglu memory surface stated in the stress space. The introduction of a new internal variable in the form of virtual back-stress, which characterizes the behavior of the material in the case of a large strain amplitude, significantly reduced the number of material constants. Moreover, these parameters were now relatively easy to identify. Particular effects of cyclic plasticity could be described thanks to the introduction of dependency between selected parameters of the Chaboche kinematic hardening rule and the nonlinear isotropic hardening rule and the radius of memory surface $R_M$. The model contained 23 parameters in total, two of which were considered as zero for the SS304 material used here. The number of required parameters was less than one-third of the number required for the Kang model [15], while maintaining the accuracy of the description of the stress–strain behavior. Acceptable results were also obtained by the new cyclic plasticity model in simulations of our own experimental data for austenitic steel 08Ch18N10T. The idea of a stress-based memory surface applied to a virtual back-stress can also be used with other nonlinear kinematic hardening rules. The authors will focus on incorporating the modified Abdel-Karim–Ohno hardening rule [8] into the proposed model to get a better prediction of the ratcheting and stress relaxation of stainless steels in future works. Some interesting results of the Abdel-Karim–Ohno model enhanced by a memory surface can be found in [28].

**Author Contributions:** R.H.: proposed the cyclic plasticity theory; J.F.: application to 08Ch18N10T stainless steel and implementation into ABAQUS; P.G.: consistent tangent modulus determination for efficient implementation into FE codes; T.K.: calibration of the model with memory to SS304 experimental data; A.M.: implementation in ANSYS.

**Funding:** The paper has been done in connection with project Innovative and additive manufacturing technology—new technological solutions for 3D printing of metals and composite materials, reg. no. CZ.02.1.01/0.0/0.0/17_049/0008407 financed by Structural Founds of Europe Union and with the project No. GA19-03282S financed by the Grant Agency of the Czech Republic (GACR). This work was also supported by the ESIF, EU Operational Programme Research, Development and Education, and from the Center of Advanced Aerospace Technology (CZ.02.1.01/0.0/0.0/16_019/0000826), Faculty of Mechanical Engineering, Czech Technical University in Prague.

**Conflicts of Interest:** The authors declare no conflict of interest.

## Abbreviations

The following abbreviations are used in this manuscript:

IDF identification specimen series
NPP nuclear power plant
SHL selected hysteresis loop
SHLs selected hysteresis loops

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
