# Peer review of "Modeling the Strain-Range Dependent Cyclic Hardening of SS304 and 08Ch18N10T Stainless Steel with a Memory Surface"

_metals, doi:10.3390/met9080832_

Round 1

Reviewer 1 Report

The paper presents a modified cyclic plasticity mode suitable for predicting material behavior of austenitic steel.

Before manuscript publication, the Authors should consider the following major comments:

•    English should be checked by a native speaker.

•    The authors should use a unique style of definitions in the whole text otherwise a reader can be confused (e.g. back-stress in Line 3 and backstress in Line123)

•    Lines: 112-113 definition of the σy is missing

•    Paragraph 2.2 has a name Virtual backstress, but the authors basically present the Chaboche model. It is not clear a difference between so-called virtual backstress and the backstress used in the Chaboche model and its significance in the paper, explain better.

•    Definition of the Rm parameter is missing in the text (Line 133).

•    Meaning of (13) is not clear in Figure 1. Probably it presents equation (13), however, it should be explained better.

•    Line 147-151: definition of ω is missing.

•    The authors talk about the estimation of Young modulus but the adopted procedure is not described. Since a precise estimation of static parameters (Young modulus and initial yield stress) is very important and influence on obtained overall results, it will be good to briefly describe the adopted procedure in the text.

•    Lines 183-185 and 203-207: Generally, Chaboche material parameters are needed to estimated using several stabilized stress-strain hysteresis loop obtained for several strain ranges. Material parameters obtained in this way are appropriate to be used for wider strain range. In case that material parameters are estimated using a single stabilized hysteresis loop (in this case 6%), estimated parameters are appropriate to use for particular strain range that has been used during the identification procedure. Therefore, it is expected that the fitting is not correct when is applied and compared considering different strain range (in this case 1%). Furthermore, different equations are suggested to use for estimation based on a single or based on several hysteresis loops. Initial starting point during estimation of parameters is another quite important aspect because it influences on the correctness of obtained parameters. How Authors choose initial starting points? During the estimation procedure, only a single strain range is considered or the parameters are estimated based on different strain ranges, justify and explain the decision?

•    Paragraph 3.3. Eq. (24) was used to determine parameters from a single hysteresis loop or considering several strain ranges?  It is proposed the equation (in the book of Lemaitre J. and Chaboche J. L.: Mechanics of Solid Materials, Cambridge University Press, Cambridge, 1990) to be adopted to estimate parameters considering several hysteresis loops obtained at different strain ranges.

•    Figure 2: Plastic strain should be written with a capital letter in order to be consistent.

•    Figure 4: the legend is missing

•    Figure 5 and Figure 6. Probably the idea is to present goodness of fitting, however, plotting experimental and predicted curves in two separate graphs is not the best solution. It is suggested to plot the experimental and the predicted curves in one bigger graph.

•    Figure 5 and Figure 6. The Authors present and compare results for 4 different cycle (N=1, 5, 10 and 30). However, Figure 4 shows that considering strain rage of 1% and 6% authors have experimental data up to the 100th /≈80th cycle, respectively. It will be nicer to presents comparison also for last cycles N=100 or 80.

•    Figure 7. Authors say that “the shapes of the hysteresis loops have very good conformity with the experimental results”. How the authors can confirm their statement because written in this way it seems quite a subjective description?

•    Estimation procedure of the yield stress is defined by the authors or is taken from other works? How the authors estimate the yield stress from the cyclic stress-strain loop? As it is written earlier, description of the estimation procedure is missing in the text and since yield stress is important parameters especially during the determination of parameters for cyclic plasticity models, it would be good to write a little paragraph in the paper with the explanation. Furthermore, why the RMSE is set to 8; an explanation is missing in the text.

•    Figure 10. Results presented in this way are not clear because some colors appear 2 times, the Authors , for example,  can adopt different styles of lines. It is suggested to remake the figure.

•    Figure 11. A legend is missing in both graphs.

•    Figure 12 – 28:

        1) all figures should have the same style of axis (Fig. 12- 15 distinguish from Fig. 16-28).

        2) The scale of the amplitude of force should be smaller (e.g. Fig. 12. from 8000 to 9000) in     order to see fitting between experimental and simulated results. It is quite hard to distinguish and see the obtained results.

        3) The scale of Errors should be consistent in all figures. For the reader, it is quite confusing and hard to understand which specimens show the best fitting considering the results presented in this way.

The Reviewer suggests to modify figures 12-28.

Reviewer 2 Report

The authors present the use of a cyclic plasticity constitutive model to describe the low-cycle fatigue behavior of austenitic stainless steels. Writing in the paper is relatively clear. The model is implemented correctly and is successful in capturing the cyclic behavior of the stainless steels, but which portions of the modeling being ‘new’ are unclear. The following suggestions are required prior to publication:

·         Use proper chemical notation for elements, 08Ch18N10T should be 8Cr18Ni10Ti

·         Explicitly make clear in the introduction what the novel contribution of the work is

·         It is unclear what makes these backstresses in this work used ‘virtual backstresses’ in comparison to the work of Jiang and Sehitgolu.

·         The labeling of the backstress as alpha_stab does not make much sense as the memory surface is able to expand with cycling, material becomes cyclically stables once Rm stops evolving

·         The paper needs more information regarding the finite element implementation in Ansys. Are the simulations single or multi-element? How many elements? Sample geometry?

·         Adjust Figures 5 and 6 so that experiment and prediction are on the same plot for comparison. Have 1, 5, 10, and 30 cycles on different sub plots

·         Section 4.1 needs to include the experimental sample geometry.

·         The inclusion of Figures 12-28 is unnecessary. These figures need to be condensed into 1 or 2 plots that provide the information about the error with increasing cycles.

·         There is significant amount of spread in the model error across samples 1-17 in 4.2 ranging from <5% to almost 20%. This should be discussed.

Round 2

Reviewer 1 Report

Earlier Reviewer’s comment

1) The authors talk about the estimation of Young modulus but the adopted procedure is not described. Since a precise estimation of static parameters (Young modulus and initial yield stress) is very important and influence on obtained overall results, it will be good to briefly describe the adopted procedure in the text.

Reply: Thank you.  This new sentence describes the way of E and σy estimation: The Young modulus and the tensile yield stress are determined from the uniaxial tensile test.

Authors replied to Reviewer that the tensile yield stress was estimated from the tensile test. However, in the texts was added an explanation that the value of the tensile yield stress was in fact chosen (not estimated) in order to obtain a good description.  

2) Figure 6. Comparison between experimental and prediction for N=30 is missing.

3) Figure 7. Authors say that “the shapes of the hysteresis loops have very good conformity with the experimental results”. How the authors can confirm their statement because written in this way it seems quite a subjective description?

Reply: Thank you. The sentences have been written in other words to describe the results of prediction more precisely.

The Reviewer noticed that this sentence was deleted while Authors wrote that it was rewritten. It would be nice to show a relative difference or error in % obtained between experimental and predicted curves.
